# Diagnosis of Autism Spectrum Disorder (ASD) by Dynamic Functional Connectivity Using GNN-LSTM

**DOI:** 10.3390/s25010156

**Published:** 2024-12-30

**Authors:** Jun Tang, Jie Chen, Miaojun Hu, Yao Hu, Zixi Zhang, Liuming Xiao

**Affiliations:** 1School of Educational Sciences, Hunan Normal University, Changsha 410081, China; zsb@hunnu.edu.cn (J.T.); jiechen@hunnu.edu.cn (J.C.); 2College of Information Science and Engineering, Hunan Normal University, Changsha 410081, China; hu@hunnu.edu.cn (M.H.);

**Keywords:** autism spectrum disorder (ASD), dynamic functional connectivity, graph neural networks (GNNs), long short-term memory (LSTM)

## Abstract

Early detection of autism spectrum disorder (ASD) is particularly important given its insidious qualities and the high cost of the diagnostic process. Currently, static functional connectivity studies have achieved significant results in the field of ASD detection. However, with the deepening of clinical research, more and more evidence suggests that dynamic functional connectivity analysis can more comprehensively reveal the complex and variable characteristics of brain networks and their underlying mechanisms, thus providing more solid scientific support for computer-aided diagnosis of ASD. To overcome the lack of time-scale information in static functional connectivity analysis, in this paper, we proposes an innovative GNN-LSTM model, which combines the advantages of long short-term memory (LSTM) and graph neural networks (GNNs). The model captures the spatial features in fMRI data by GNN and aggregates the temporal information of dynamic functional connectivity using LSTM to generate a more comprehensive spatio-temporal feature representation of fMRI data. Further, a dynamic graph pooling method is proposed to extract the final node representations from the dynamic graph representations for classification tasks. To address the variable dependence of dynamic feature connectivity on time scales, the model introduces a jump connection mechanism to enhance information extraction between internal units and capture features at different time scales. The model achieves remarkable results on the ABIDE dataset, with accuracies of 80.4% on the ABIDE I and 79.63% on the ABIDE II, which strongly demonstrates the effectiveness and potential of the model for ASD detection. This study not only provides new perspectives and methods for computer-aided diagnosis of ASD but also provides useful references for research in related fields.

## 1. Introduction

Autism spectrum disorder (ASD) is a neurodevelopmental disorder whose core features include significant social communication difficulties, a very narrow range of interests, repetitive and stereotyped behaviors, and abnormal perceptual responses. The symptoms of ASD are typically evident in early childhood, but social deficits may not be noticeable in complex social environments. However, these deficits can be observed in settings involving intricate social interactions. In addition, the diagnosis of ASD is a complex process that involves a series of meticulous steps including, but not limited to, an initial assessment by a caregiver, a professional interview with a physician, long-term clinical monitoring, and a comprehensive evaluation. This process is not only time-consuming and laborious but also increases the economic cost and technical difficulty of diagnosis. Therefore, researchers need to develop new technologies to achieve efficient identification of ASD and streamline its early screening and diagnostic process. This will not only improve the efficiency of ASD diagnosis and reduce the burden on the healthcare system but also provide more timely and targeted interventions for children with ASD, thereby improving their quality of life.

Static functional connectivity (SFC) assumes that the brain’s functional connectivity (FC) is constant during the sampling period when analyzing fMRI data, ignoring the dynamic changes of FC on the time scale. However, numerous studies have shown that the brain’s FC changes dynamically in the resting state [1]. DFC describes not only the spatial variation of FC states but also the variation of FC states over time. This spatio-temporal complexity can provide richer information for the study of the human brain but at the same time increases the difficulty of feature extraction. In recent years, there have been more and more studies on DFC, aiming to understand the relationship between ASD and FC dynamics, as well as to identify biomarkers based on DFC. Zhao et al. [2] extracted higher-order dynamically relevant features from DFC based on the idea of “correlation of correlation” and introduced the central moment method to solve the problem of dynamic mismatch of DFC between subjects, to extract the dynamic mismatch of DFC and FC among the subjects, and to extract the dynamic mismatch of DFC. Liu et al. [3] used an improved multi-task feature selection method, combining elastic networks and stream regularization, to filter discriminative features from DFCs and found that functional connectivity related to the left thalamus, right precuneus, and left insula is valuable in ASD diagnosis. Yan et al. [4] designed a fully connectivity bidirectional long short-term memory network (Full-BiLSTM), using the functional connectivity of each frame of the DFC as input, and trained the Full-BiLSTM to learn the past and future information of each window of the DFC. Their study achieved 73.6% accuracy on the Alzheimer’s Disease Neuroimaging Initiative dataset (ADNI), which demonstrates the superior processing capability of LSTM for temporal DFC. Although these studies have successfully utilized DFCs to explore the pathogenesis of ASD, they usually treat DFCs as matrices to extract features directly, which may ignore the structural details of higher-order brain networks.

The spatio-temporal information embedded in DFC is very rich; however, most of the previous studies are limited to dealing with temporal or spatial information individually, and there is a lack of a unified model that can deal with spatio-temporal information simultaneously. To address this problem, Chen et al. [5] proposed a novel diagnostic strategy for ASD utilizing spatio-temporal deep neural networks with an integrated self-attention mechanism. Specifically, they used the sliding window method to construct the DFC in the time dimension and applied Kendall’s rank correlation coefficient to extract features. Subsequently, a deep neural network based on a multi-head self-attention mechanism was designed to extract more discriminative spatial and temporal high-level abstract features. The strategy was systematically experimented on a large-scale ASD dataset and demonstrated better performance than existing models in both tenfold and inter-site cross-validation. Hu et al. [6] proposed a method for constructing DFCs based on a multi-scale sliding window and a multi-scale convolution neural network (MsCNN) to learn and analyze DFCs at various scales. They performed a deep fusion of features learned at different scales for the diagnostic classification of brain diseases. The study was validated on several datasets, including the ADNI and ABIDE datasets, and both achieved satisfactory classification accuracies. Xing et al. [7] designed a spectral convolution-based LSTM layer to process DBNs. Spectral convolution was used to aggregate the information from neighboring regions of the brain area, while the LSTM captured the temporal information of DBNs through multiple contiguous units. This approach effectively combines spatial and temporal information and provides new perspectives for understanding the dynamic functional organization of the brain. Gadgil et al. [8] developed a spatio-temporal graph convolutional network (ST-GCN) that uses 1D convolution kernels to combine functional connectivity and temporal characterization in the BOLD signals. Deng et al. [9] used a bi-directional LSTM coupled with a Transformer to process fMRI data and achieved 70.26% accuracy in the task of recognizing ASD. Deng et al. [10] then used a spatio-temporal Transformer to extract spatio-temporal features of fMRI for classification. Azevedo et al. [11] proposed a spatio-temporal learning module, which consists of spatial embedding learning for the GNN and TCN for temporal embedding learning. Unlike ST-GCN, the input is the entire BOLD signal, and TCN is used to capture signal features. These methods emphasize the importance of spatio-temporal information in understanding brain function. Wang et al. [12] proposed a DFC-based end-to-end temporal dynamic learning (TDL) method for brain disease diagnosis. They first used overlapping sliding windows to convert rs-fMRI time series into DFCs and then introduced a group fusion Lasso regularizer to capture the global temporal dynamics of these networks. However, the main limitation of TDL is that it only models linear relationships of dynamic brain networks and does not use nonlinear functions to model correlations between brain regions or correlations between time-map structures, which limits its ability to capture complex nonlinear dynamics and information at multiple time scales. To overcome this limitation, Lin et al. [13] proposed a convolutional recurrent neural network (CRNN), which utilizes convolution to construct a brain network and then extracts sequence features via LSTM. This approach combines the advantages of convolutional neural networks in spatial feature extraction and the expertise of LSTM in time series analysis, providing a new tool for capturing the complex dynamics of brain networks.

However, the above-mentioned methods only consider the fixed-scale temporal dependence problem when using LSTM for DFC and do not extract the variable time-scale dependence problem of FCs between different time windows in DFC. In order to address the lack of FC in ASD diagnosis in the analysis of SFC in terms of time-scale information extraction, in this paper, we propose the GNN-LSTM model. The model achieves a comprehensive and efficient analysis of DFCs by fusing the respective advantages of GNN and LSTM and combining jump connections with design-specific pooling operations. The main contributions of this paper are summarized as follows:Aiming at the problem that SFC cannot effectively capture the change of FC over time, in this paper, we use the sliding window method to compute the DFC. This method not only can compute the time series into multiple FCs but also provides a rich and comprehensive database for the subsequent feature extraction and analysis.To extract representative node features from dynamic brain networks, an innovative pooling operation, dynamic graph pooling (DG-Pool), derives the final node feature representations by synthesizing the graphical representations of each time window, effectively preserving the key information of the time scale, and providing a powerful feature support for the ASD detection provides strong feature support.To address the problem of DFC dependence on time scales, we introduce a jump connection between GNN-LSTM units. This connection mechanism allows the model to retain higher-order time-scale information, thus realizing the accurate extraction of DFC time-space features. This improvement not only enhances the expressive capability of the model but also strengthens its ability to handle complex dynamic data.Experimental validation on the ABIDE dataset demonstrates that the GNN-LSTM model exhibits high efficiency in detecting ASD. The results show that the model excels in processing DFC, offering new insights for ASD diagnosis and treatment techniques and paving the way for in-depth research on the pathological causes of ASD. These findings are crucial for enhancing the accuracy of ASD diagnosis, alleviating the burden on the healthcare system, and providing timely and effective interventions for children with ASD.

## 2. Materials and Methods

### 2.1. Data Pre-Processing

Prior studies have primarily focused on static functional connectivity analyses to explore the pathological mechanisms of ASD. However, static analyses tend to overlook the time-varying nature of brain activity and fail to capture crucial information over time. While static functional connectivity analysis has provided some insight into ASD research, its limitations are becoming increasingly evident. DFC analysis has the potential to comprehensively capture the time-varying properties of brain activity, revealing dynamic differences in brain connectivity patterns between ASD patients and typically developing individuals. These differences can offer more accurate insights into ASD and are expected to improve diagnostic precision. To capture the relevant features of functional connectivity over time, we employ the sliding window method to calculate DFC. Initially, the time series is segmented into overlapping windows using a sliding window of size L=40. The window size slides in steps of 10 time points, generating a total of $N_{win}=14$ windows for each sample. The FC of each window is then estimated using the Ledoit–Wolf (LDW) regularized shrinkage estimator. The LDW covariance estimator is calculated as described in Equations (1)–(3).
(1)$$\tilde{\Sigma }=\left(1-\alpha \right)\hat{\Sigma }+\alpha \Delta$$
(2)$$\Delta =\left(Tr\left(\hat{\Sigma }\right)/N\right)I_{N}$$
(3)$$\hat{\Sigma }=\frac{1}{T}\sum_{t=1}^{T}\left(Y_{t}-\bar{Y}\right){\left(Y_{t}-\bar{Y}\right)}^{T}$$
where α is the shrinkage parameter, $I_{N}$ is the identity matrix of N×N, *N* denotes the number of ROIs, $Tr\left(\cdot \right)$ denotes the trace of the matrix, $\hat{\Sigma }$ is the sample covariance matrix of N×N, T denotes the length of the time series in each window, and the sample mean $\bar{Y}$ is calculated as in Equation (Equation 4).
(4)$$\bar{Y}=\frac{1}{T}\sum_{t=1}^{T}Y_{t}.$$
The correlation matrix is calculated as in Equation (Equation 5).
(5)$$R=D^{-1/2}\tilde{\Sigma }D^{-1/2}$$
where *D* denotes the diagonal matrix of $\tilde{\Sigma }$.

The DFC is formed by overlaying the SFC across the time series. Once the DFC is computed, each sample can be visualized as an undirected graph $G_{t}=\left(V,E,\Omega \right)$ at time *t*, like the SFC. However, unlike the SFC, the node features of the samples at time *t* are derived using the LDW regularized shrinkage estimator. The flowchart for computing the DFC is shown in Figure 1.

### 2.2. Methods

In this paper, we introduce a GNN-LSTM model, outlined in Figure 2. The model utilizes the sliding window method to construct the DFC from fMRI data, facilitating feature extraction on both temporal and spatial scales. To achieve this, the model integrates graph convolution operations into the original LSTM cells. Following feature extraction, the model utilizes a DG-Pool to effectively pool the final node representations from the graph for classification. To address the DFC’s time scale dependency, the model incorporates a jump connection mechanism between the cells internally. Lastly, the model merges the higher-order features of the DFC with the lower-order features of the time series by feeding them into the fully connected layer to obtain the classification results.

After fMRI computes the DFC via the method shown in Section 2.1, each sample can be represented as $D=\{G_{1},\ldots ,G_{T}\}$, where *T* denotes the number of windows in the computation of the DFC, and the samples can be represented as an undirected graph $G_{t}=\left(V,E_{t},\Omega_{t}\right)$ at the moment *t*. Similarly, $G_{t}$ has an associated node feature set $E_{t}=\left[\varepsilon_{1},\ldots ,\varepsilon_{n}\right]$. As a model for the classification task, it is necessary to learn a mapping function $f:{\left\{G_{t}\right\}}_{t=1}^{T}\to \hat{y}$.

To fully extract the spatio-temporal features of brain activity, this chapter uses LSTM to act directly on the data of the original fMRI time series to obtain the low-order features, a process that can be expressed by Equation (Equation 6).
(6)$$\hat{S}=LSTM\left(\left\{S\right\}\right)$$
where S denotes the fMRI time series data obtained after processing through the CPAC pipeline.

LSTM is a feedback deep learning architecture proposed to solve the problem of poor learning ability of recurrent neural networks (RNNs) in longer sequences due to the gradient vanishing problem. A standard LSTM unit consists of a cell ($C_{t}$), an input gate ($I_{t}$), an output gate ($O_{t}$), and a forget gate ($F_{t}$). However, there are two problems with traditional LSTM:The fully connected operator in LSTM ignores spatial correlation.Fixed jump lengths within LSTMs are constrained by the inability to utilize variable length dependencies.

To overcome these limitations, in this paper, we propose a novel GNN-LSTM model that combines the advantages of GNN and LSTM to model temporal features at the graph level. Specifically, the GNN-LSTM module effectively models temporal representations in dynamic brain networks by incorporating graph convolution operations. The module contains input gates, oblivion gates, and output gates and directly takes DFC as input. In GNN-LSTM, the LSTM component is responsible for learning the temporal embedding of all the brain function graphs in the DFC, while the GCN is used to capture the graph structural properties of the nodes and their interrelationships in each brain function graph. Thus, the operation of each gate is realized by stacking graph convolutional layers. In addition, to address the variable dependence of time scale in DFC, the model employs a jump connection mechanism to capture multi-level information on the DFC time scale by adjusting the jump step size. Notably, the inputs, hidden states, and cell memories of the GNN-LSTM cells use graph structures instead of vector structures in traditional LSTMs. This design allows the model to better handle data with complex spatial and temporal dependencies. The GNN-LSTM cell contains three inputs: $H_{t-p}$, $G_{t}$, and $\varepsilon_{t}^{G}$, where $H_{t-p}$ denotes the hidden state obtained at the t−p step, *p* is the number of hidden units skipped, $G_{t}$ is the graph in the sequence of dynamic graphs at the *t* moment, and $\varepsilon_{t}^{G}$ is the corresponding feature of the *t*th graph in the sequence of input graphs. The inputs, hidden states, and cell memories of GNN-LSTM are all graph structures instead of vector structures as in traditional LSTM. The update process is defined as the Equations (7)–(12).
(7)$$\mathrm{Input}\;\mathrm{gate}:\;I_{t}=\mathrm{Sigmoid}\left(W_{i}*\mathrm{Gconv}\left(G_{t}\right)+{\hat{W}}_{i}*\mathrm{Gconv}\left(H_{t-p}\right)\right)+b_{i}$$
(8)$$\mathrm{Forget}\;\mathrm{gate}:\;F_{t}=\mathrm{Sigmoid}\left(W_{f}*\mathrm{Gconv}\left(G_{t}\right)+{\hat{W}}_{f}*\mathrm{Gconv}\left(H_{t-p}\right)\right)+b_{f}$$
(9)$$\mathrm{Output}\;\mathrm{gate}:\;O_{t}=\mathrm{Sigmoid}\left(W_{o}*\mathrm{Gconv}\left(G_{t}\right)+{\hat{W}}_{o}*\mathrm{Gconv}\left(H_{t-p}\right)\right)+b_{o}$$
(10)$$\mathrm{Input}\;\mathrm{modulation}:\;U_{t}=\mathrm{ReLU}\left(W_{c}*Gconv\left(G_{t}\right)+{\hat{W}}_{c}*Gconv\left(H_{t-p}\right)\right)+b_{c}$$
(11)$$\mathrm{Cell}\;\mathrm{memory}:\;C_{t}=Tanh\left(I_{t}*U_{t}+F_{t}*C_{t-p}\right)$$
(12)$$\mathrm{Output}:\;H_{t}=O_{t}*Tanh\left(C_{t}\right)$$
where $I_{t}$, $F_{t}$, $O_{t}$, $U_{t}$, $H_{t}$, and $C_{t}$ denote input gates, forget gates, output gates, modulated inputs, hidden states, and cell memories, respectively, and they are all graph-structured data. $W_{i}$, $W_{f}$, $W_{o}$, $W_{c}$, ${\hat{W}}_{i}$, ${\hat{W}}_{f}$, ${\hat{W}}_{o}$, and ${\hat{W}}_{c}$ denote the trainable weights of the different gates, and $b_{i}$, $b_{f}$, $b_{o}$, and $b_{c}$ denote their bias coefficients, respectively. The update operation of the graph convolution operation Gconv on the nodes at layer *l* can be represented by Equation (Equation 13)
(13)$$E_{t}^{G^{\left(l+1\right)}}=Gconv\left(G_{t}\right)=relu\left(E_{t}E_{t}^{G^{\left(l\right)}}\Omega_{t}W_{G}^{\left(l\right)}\right)$$
where $W_{G}^{\left(l\right)}$ denotes the *l*th layer graph convolution trainable parameter matrix, $E_{t}$ and $\Omega_{t}$ denote the set of edges and the set of weights of the edges of the graph at the moment *t*, respectively, and $E_{t}^{G^{\left(l+1\right)}}$ denotes the node features that have been convolved after the *l* step of the graph convolution after the *l* step graph convolution.

The GNN-LSTM model can capture long-term dependencies with variable lengths by adjustable jump lengths. This is shown in Figure 3.

When p=1, output ${\hat{H}}_{T}^{\left(1\right)}=W_{T}^{\left(1\right)}H_{T}^{\left(1\right)}$, when p=2, output ${\hat{H}}_{T}^{\left(2\right)}=W_{T}^{\left(2\right)}H_{T}^{\left(2\right)}+W_{T-1}^{\left(2\right)}H_{T-1}^{\left(2\right)}$, when p=3, output ${\hat{H}}_{T}^{\left(3\right)}=W_{T}^{\left(3\right)}H_{T}^{\left(3\right)}+W_{T-1}^{\left(3\right)}H_{T-1}^{\left(3\right)}+W_{T-2}^{\left(3\right)}H_{T-2}^{\left(3\right)}$, and so on. Where $H_{T}^{\left(1\right)}$, $H_{T}^{\left(2\right)}$, $H_{T}^{\left(3\right)}$ denote the output of step *T* of the model. We use a linear transformation to combine the outputs of multiple jump connections, which can be represented by Equation (Equation 14).
(14)$${\hat{H}}_{T}^{C}=\sum_{p=1}^{P}\sum_{i=T}^{T-p+1}W_{i}^{\left(p\right)}H_{i}^{\left(p\right)}+b$$
where $H_{i}^{\left(1\right)}$, $H_{i}^{\left(2\right)}$, …, $H_{i}^{\left(p\right)}$ is the output with multiple connections and $W^{\left(p\right)}$ is the trainable parameter.

Considering that the jump-joining mechanism may cause the model to “memorize” rather than generalize the patterns in the DFC data, we add Dropout between the GNN-LSTM cells.

Since there are multiple dynamic brain graphs in DFC, how to select the graph with the highest contribution to classification requires the design of a new pooling layer, DG-Pool, and the methods of Cangea et al. [14] and Gao et al. [15] are adopted for pooling operations. Specifically, the model decides whether to remove a node from the pool by projecting the node features onto the learnable vector, which is computed as in Equations (15)–(19).
(15)$$s^{\left(l\right)}={\hat{H}}^{\left(l+1\right)}w^{\left(l\right)}/{\left\|w^{\left(l\right)}\right\|}_{2}$$
(16)$${\tilde{s}}^{\left(l\right)}=\left(s^{\left(l\right)}-\mu \left(s^{\left(l\right)}\right)\right)/\sigma \left(s^{\left(l\right)}\right)$$
(17)$$i=\mathrm{top}k\left({\tilde{s}}^{\left(l\right)},k\right)$$
(18)$$H^{\left(l+1\right)}={\left({\hat{H}}^{\left(l+1\right)}⊙Sigmoid\left({\tilde{s}}^{\left(l\right)}\right)\right)}_{i}$$
(19)$$E^{\left(l+1\right)}=E_{i,i}^{\left(l\right)}$$
where ${\left\|\cdot \right\|}_{2}$ denotes the L2-norm, μ and σ denote the mean and variance of the input and output vectors, $topk\left(\cdot \right)$ denotes the index of the largest *k* vectors to be found in ${\tilde{s}}^{\left(l\right)}$, ⊙ denotes the multiplication of the corresponding elements of the two matrices, and ${\left(\cdot \right)}_{i,j}$ means to find the elements of the *i*th row, *j*th column.

DG-Pool uses $Sigmoid({\tilde{s}}_{m}^{\left(l\right)})$ to convert ${\hat{s}}_{m}^{\left(l\right)}=\left[{\hat{s}}_{m,1}^{\left(l\right)},\ldots ,{\hat{s}}_{m,N^{\left(l\right)}}^{\left(l\right)}\right]$ in descending order to select the features of the nodes that need to be left behind, where the *m* subscript denotes the *m*th sample and *N* denotes the total number of nodes in the sample. Meanwhile, so that the pooling layer of the model can better capture the time dependence in the DFC features, based on the concept of the cross-entropy loss function, the pooling loss $L_{Pool}$ is designed in this chapter, and the computation can be expressed by Equation (Equation 20).
(20)$$L_{Pool}^{\left(l\right)}=-\frac{1}{M}\sum_{m=1}^{M}\frac{1}{N^{\left(l\right)}}\left(\sum_{i=1}^{k}log\left({\hat{s}}_{m,i}^{\left(l\right)}\right)\right)+\sum_{i=1}^{N^{\left(l\right)}-k}log\left(1-{\hat{s}}_{m,i+k}^{\left(l\right)}\right)$$
where *l* denotes the *l*th pooling layer and *k* is a hyperparameter which indicates how many nodes will be left in the pooling layer.

After extracting the lower-order features and higher-order graph features of the time series, the model splices these two types of features to form a composite vector representation containing spatio-temporal information. This composite vector is then fed into the fully connected layer to generate the predicted labeled values of the samples. By comparing with the true label values, the model calculates the loss $L_{ce}$ for the classification task. In addition to the classification loss, the model also considers a pooling loss $L_{Pool}$, which is a loss function designed to optimize the graph pooling process and improve the efficiency of extracting graph structure information. The combined loss function L of the model is the sum of the classification loss $L_{ce}$ and the pooling loss $L_{Pool}$, which is used to guide the optimization direction of the model throughout the training process. It can be expressed by Equation (Equation 21).
(21)$$L=L_{ce}+\sum_{l=1}^{L}L_{Pool}^{\left(l\right)}.$$
By minimizing the integrated loss function *L*, the model can simultaneously optimize its performance on the extraction of spatio-temporal features, graph structure information, and classification tasks, resulting in more accurate prediction and more efficient feature learning.

## 3. Experiments and Results

### 3.1. Experimental Setup

In this paper, we validated the GNN-LSTM model on the ABIDE dataset. It is worth noting that the source of the ABIDE dataset covers several different sites, resulting in significant differences in the lengths of data series produced at different sites. This discrepancy is particularly evident when calculating the DFC using the sliding window method, as a given window size and displacement size will result in a varying number of correlation matrices being generated for time series of different lengths. In essence, the window size and window displacement size together define the size of the receptive field when computing the FC, similar to the convolutional kernel in a convolutional neural network.

To standardize the number of FCs in the DFC for subsequent analysis and comparison, we employed a screening strategy in this chapter. Specifically, we selected only samples of consistent length for the experiment to ensure data consistency and comparability. This approach establishes a solid foundation for the subsequent experimental analysis.

Regarding the experimental hardware environment, the hardware specifications used are detailed in Table 1. These high-performance hardware resources ensure the smooth running of the experiments and provide sufficient computational support for model training and evaluation. With reasonable hardware configuration, we completed the training and testing of the model and verified the effectiveness and superiority of the GNN-LSTM model in the field of ASD detection.

To guarantee the generalization ability and effectiveness of the model, the dataset was randomly partitioned into a training set and a test set based on the standard 8:2 ratio. To mitigate the overfitting phenomenon that may occur during model training, we adopt the early stopping method as an optimization strategy. This method continuously monitors the performance of the model on independent validation sets and triggers the stopping mechanism once the model fails to show significant performance improvement on the validation sets in consecutive training epochs, thus effectively avoiding overfitting on the training data and guaranteeing its generalization ability on the unseen data. In addition, to regulate the training process of the model more finely, this study also introduces a dynamic learning rate strategy. This strategy allows the learning rate to be adaptively adjusted according to predefined rules and strategies at different stages of training to match the specific needs of the model at different learning stages. This approach aims to balance the training amplitude of the model more effectively and further improve the performance and stability of the model. All the details of the parameters involved in the experiments have been exhaustively listed and described in Table 2.

### 3.2. Statistical Metrics

The task is to detect ASD patients, which is a binary classification task, and the commonly used evaluation metrics are accuracy (ACC), sensitivity (SEN), precision (PRE), and F1_score, and these metrics are computed as in Equations (22)–(25).
(22)$$\mathrm{ACC}=\frac{TP+TN}{TP+FP+TN+FN}$$


(23)
$$\mathrm{SEN}=RECALL=\frac{TP}{TP+FN}$$



(24)
$$\mathrm{PRE}=\frac{TP}{TP+FP}$$



(25)
$$F1\_\mathrm{score}=\frac{2*PRE*SEN}{PRE+SEN}$$


In the above formula, TP denotes positive samples predicted for positive categories, FP denotes negative samples predicted for positive categories, TN denotes negative samples predicted for negative categories, and FN denotes positive samples predicted for negative categories. Accuracy indicates the ratio of the number of samples correctly predicted by the model to the total number of samples, and it is a basic indicator of the effectiveness of a model. Sensitivity, recall expresses the proportion of positive samples out of the total positive samples predicted by the model, and it is often used to measure a model’s ability to objectively recognize samples in positive categories. Precision, on the other hand, is a good expression of a model’s subjective predictive ability, and it indicates the proportion of samples predicted by the model to be in the positive category that are actually in the positive category. Precision and recall illustrate the model’s performance of predicting positive classes from two perspectives, and to synthesize these two evaluation metrics, researchers usually use F1_score to evaluate the model, which integrates the model’s prediction performance in both subjective and objective ways. As can be seen from Equation (Equation 25), the larger the TP is, the larger the F1_score is, and the F1_score integrally reflects the accuracy of the model. This paper also uses the area under the curve (AUC) enclosed with the coordinate axis as an evaluation index, which is obtained by calculating the area under the ROC curve.

### 3.3. Comparison with State-of-the-Art (SOTA) Models

As shown in Table 3 and Table 4, the experimental results of the GNN-LSTM model on the ABIDE I and ABIDE II datasets fully demonstrate its superior ability to extract the temporal-space features of DFC. Compared with the recent models on SFC and DFC, the GNN-LSTM model proposed in this chapter not only exhibits excellent performance in terms of accuracy but also shows significant improvement in terms of sensitivity. This result not only validates the effectiveness of the model design but also further confirms the applicability of the strategy of combining GNN and LSTM in capturing complex spatio-temporal dynamic patterns. By combining the complementary advantages of GNN and LSTM, the model can extract spatio-temporal features from fMRI data more effectively, thus providing a more accurate diagnosis of ASD.

### 3.4. Ablation Study

To analyze the specific contribution of each module in the model to the final classification performance, ablation experiments are designed in this subsection. Not only the key components in the GNN-LSTM model (e.g., DG-Pool layer and jump connections) are studied for ablation in the experiments of this section, but also the baseline model widely recognized in the field is selected as a control. By comparing the performance of these models in terms of key evaluation metrics such as accuracy and sensitivity, the specific contribution of each module in the GNN-LSTM model to performance improvement can be identified. In addition, to visualize the model performance more intuitively, ROC curves are plotted in this section as an auxiliary analysis tool.

According to the experimental results shown in detail in Table 5 and Table 6, the performance of the GNN-LSTM model exhibits improved performance on both ABIDE I and ABIDE II datasets compared to the baseline approach. In particular, the accuracy and other performance metrics of the model are further optimized with the introduction of jump connections and the DG-Pool layer. This improvement not only validates the effectiveness of the model design but also highlights the important role of each module in enhancing the model performance.

When dealing with binary classification tasks, especially on datasets with imbalanced distribution of positive and negative samples, the AUC value is often regarded as a comprehensive and robust model performance evaluation index. As shown in Figure 4, compared with the baseline model, the GNN-LSTM model in this chapter also achieves a significant improvement in the AUC value, which further confirms its superior performance when dealing with unbalanced data.

### 3.5. Comparison of Different Pooling Methods

This section shows the effectiveness of the DG-Pool proposed in this chapter by varying different pooling methods in the model. Its impact on the model performance is illustrated by testing the accuracy and AUC of the model with different pooling methods on ABIDE I and ABIDE II, respectively.

Diffpool: Diffpool de-layers the aggregated graph nodes through a differential module to obtain the final node representation.

SAGpool: SAGpool adaptively learns the importance of nodes from the graph through graph convolution and then utilizes the TopK mechanism to obtain the final node representation.

Attpool: Attpool generates the final node representation by adaptively selecting the nodes that are important to the graph representation and by aggregating the attention-weighting information in the nodes.

As shown in Figure 5, compared with various pooling methods, the DG-Pool has better accuracy as well as AUC, and the difference between the accuracy and AUC of the model using the DG-Pool is small, which fully demonstrates that in the pooling process, the DG-Pool can effectively aggregate data according to different time windows, which ultimately leads to improved model performance.

### 3.6. Feature Visualization

This section aims to showcase the superior feature extraction ability of the GNN-LSTM model using visualization techniques. To accomplish this, we will utilize chord graphs as a feature visualization tool. Chord graphs effectively illustrate the complexity and strength of relationships between nodes. String diagrams will visually represent node interactions using arcs and strings connecting these arcs. Visual elements such as color, thickness, and position of the strings accurately reflect graph feature attributes, including connection strength, directionality, and other essential information.

In Figure 6, the graph depicts the connections between different categories of sample ROIs in DFC. ROIs with strong connections are represented by larger sectors, while those with weaker connections appear as smaller sectors. By comparing the ROI connections between the ASD and HC samples, we can discern distinct differences. In the HC sample, there is more frequent communication between brain regions numbered 42–55 compared to ASD, as evidenced by denser chords and larger sectors in the graph. Conversely, the communication pattern between brain regions numbered 90–93 is notably different in the ASD sample, revealing discrepancies in the corresponding regions in the figure. This discovery not only affirms the effectiveness of the GNN-LSTM model in feature extraction but also provides valuable insights for understanding the pathological mechanism of ASD.

### 3.7. Cluster Analysis

In this section, We use KNN clustering to analyze the DFC after model feature extraction. Based on previous studies, We cluster the entire dataset into six states that may represent the functional connectivity patterns of ASD and HC in different cognitive or affective states. These states may involve different neural network activity patterns, reflecting the complexity and diversity of the brain in processing information. Meanwhile, for each category, three states were clustered, and by comparison, it was found that the six states clustered by the former included the states of ASD and HC.

To show the difference between ASD and HC in terms of brain state transitions more intuitively, several groups of representative ASD and HC samples were selected in this section, and their state transfer diagrams were plotted separately. As shown in Figure 7 and Figure 8, these diagrams demonstrate the differences between the two groups regarding brain state transition pathways, frequency, and stability using visualization. By comparing and analyzing these state transfer diagrams, a deeper understanding of the brain functional connectivity abnormalities in ASD patients can be achieved, providing a useful reference for future research.

As shown in Figure 9, after the processing of the GNN-LSTM model, the spatio-temporal-dependent features in the DFC are effectively extracted. These features are represented in the figure as multiple states of DFC, which can be regarded as different modes of the brain when performing information processing and communication. Further analysis revealed that there was a significant difference in the probability of state transitions between ASD and HC. Specifically, patients with ASD had a nearly fourfold difference in the probability of moving from state 1 to state 3 compared to HC and a nearly eightfold difference in the likelihood of moving from state 1 to state 2 compared to HC. This finding reveals a significant difference between brain pattern transitions in ASD patients and healthy populations.

As shown in Figure 10 and Figure 11, the state transition pattern of the DFC exhibits a clear tendency that the state is usually maintained in the current state with a higher probability and transitions to other states with a lower probability. This observation is consistent with the state transfer matrix in Figure 4, Figure 5, Figure 6, Figure 7 and Figure 8. It is worth noting that although there is some commonality between ASD and HC in terms of the probabilities of state maintenance and transitions, there are also differences between them. These differences are reflected in the frequency of state transitions, the paths, and the stabilized states after the transitions. These findings suggest that an in-depth analysis of transitions between DFC state modes could help reveal more details of ASD pathogenesis.

### 3.8. Hyperparameter Discussion

In this section, as shown in Figure 12, we explore the significance of the jump number *p* in the model. The values of *p* will be sequentially set to 1, 2, 3, and 4. Subsequently, we will compare the model’s ACC and the AUC on the ABIDE dataset to ascertain the most suitable jump number.

Upon thorough examination of the ABIDE I and ABIDE II datasets, it has become apparent that the model’s accuracy and AUC were maximized when the value of *p* was set to 2. This observation implies the presence of an optimal *p* that enables the model to effectively retain temporal information, thus enhancing classification performance. A diminutive *p* may impede the model’s capacity to sufficiently capture temporal dynamics in the DFC, resulting in inadequate information extraction. Conversely, an elevated *p* may introduce extraneous noise or redundant information, consequently precipitating a decline in performance. Significantly, an excessively high *p*-value augments the number of graph convolution layers in the model, potentially leading to over-smoothing.

In this section, we delved into two crucial parameters of the sliding window method for calculating DFC: window size *L* and window step. To comprehensively assess the impact of these parameters on model performance, we will explore different parameter combinations in empirical studies. Specifically, we will set the value range of window size *L* to [40, 50, 60] and the value range of window step to [10, 20, 30].

As portrayed in Figure 13, the choice of window size and step significantly impacts the performance of the model during the DFC construction process. Notably, when the window size is set to 40 and the window step is 10, the model achieves the highest accuracy for both ABIDE I and ABIDE II datasets. This outcome is attributed to the combined effect of window size and step on the receptive field’s size in the DFC, as well as the retention of information over time. An appropriate window size enables the model to capture ample contextual information, while a window that is too small may result in an inadequate representation of data dynamics. Furthermore, the selection of the window step is also pivotal, as a suitable displacement can ensure effective feature extraction at various time scales, while an excessively large displacement may lead to information loss and impact the model’s classification performance.

## 4. Conclusions

In this paper, we provide an exhaustive description of the GNN-LSTM model, which combines GNN with LSTM for feature extraction from DFC. Our experiments on the ABIDE dataset demonstrate the effectiveness of the dynamic graph pooling operation, DG-Pool, in aggregating functional connectivity information on time series. Additionally, the model’s use of jump connections enables it to handle correlation information between FCs at different time scales, thereby enhancing its performance. The experimental results show that the model accurately captures complex patterns in DFC and distinguishes brain network pattern transitions between ASD and HC. This model serves as a valuable reference for understanding the brain network mechanism of ASD and offers new perspectives and tools for neuroscience research, as well as the development of early diagnosis and intervention strategies for related diseases.

However, there is still potential for improvement in the methodology of this paper. The pathogenesis of brain diseases such as ASD typically involves the joint influence of multiple factors, yet we only analyzed fMRI data without considering the influence of other factors such as genetics. In future studies, integrating data from more modalities, such as genetics, physiology, and environmental factors, can comprehensively analyze the causes of autism, revealing more complex pathogenesis and improving diagnostic accuracy. This approach will help in developing more effective computer-assisted diagnostic methods and improving the quality of life for patients.

## Figures and Tables

**Figure 1 sensors-25-00156-f001:**
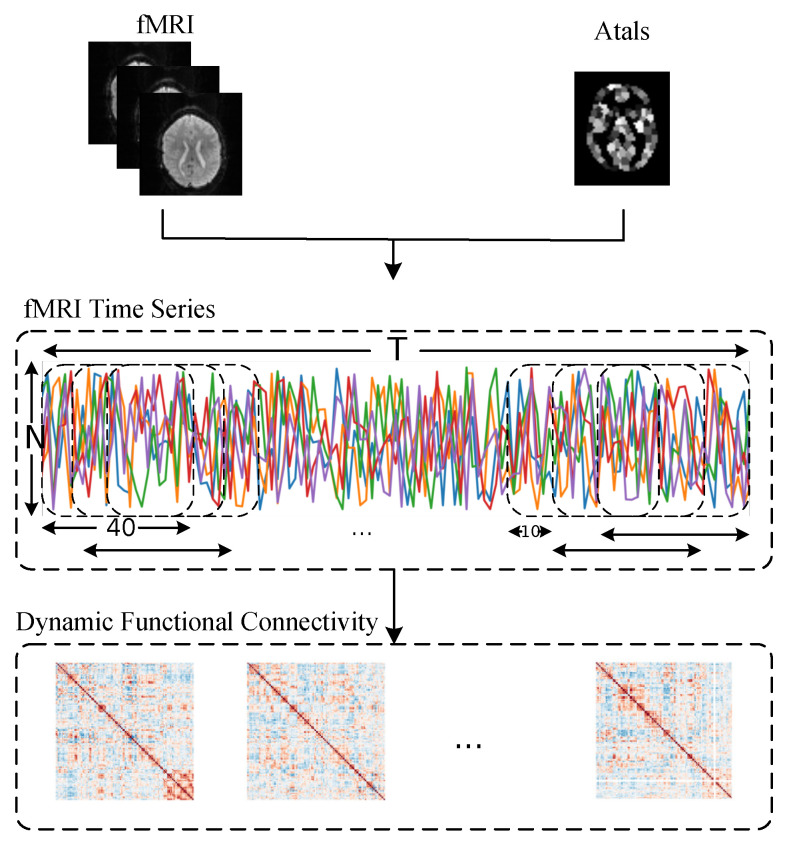
DFC calculation flow.

**Figure 2 sensors-25-00156-f002:**
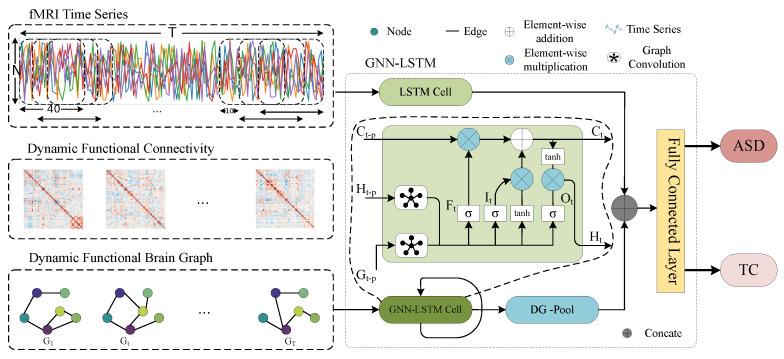
Overview of the GNN-LSTM.

**Figure 3 sensors-25-00156-f003:**
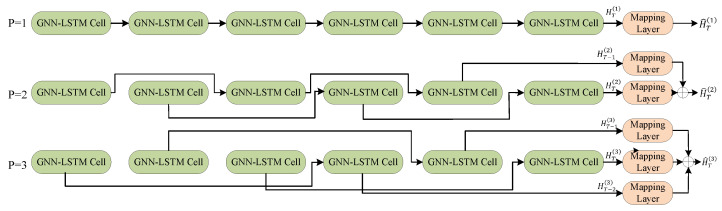
Variable length jump connection instructions.

**Figure 4 sensors-25-00156-f004:**
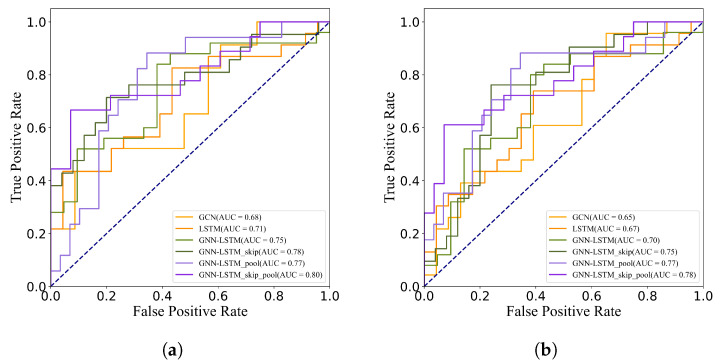
ROC curves for different methods on ABIDE. (**a**) ABIDE I. (**b**) ABIDE II.

**Figure 5 sensors-25-00156-f005:**
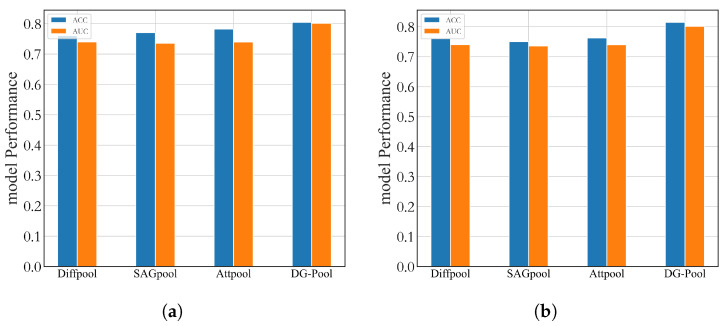
Performance of different pooling methods (**a**) ABIDE I. (**b**) ABIDE II.

**Figure 6 sensors-25-00156-f006:**
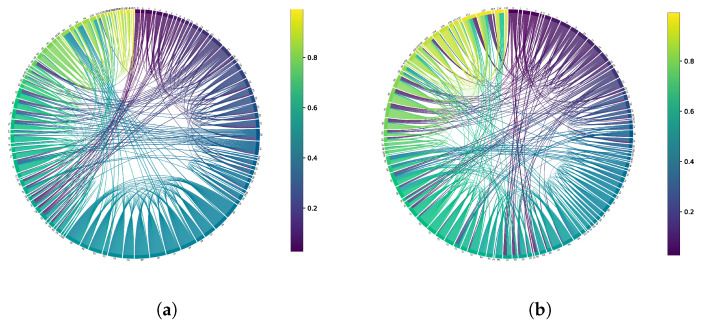
Feature matrix visualization. (**a**) HC. (**b**) ASD.

**Figure 7 sensors-25-00156-f007:**
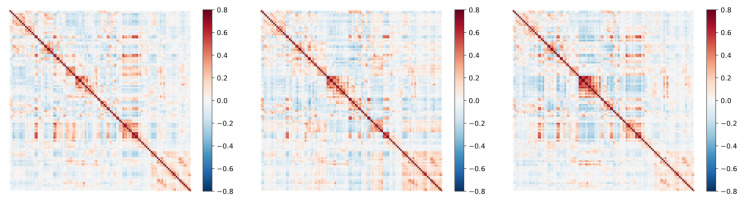
States in ASD.

**Figure 8 sensors-25-00156-f008:**
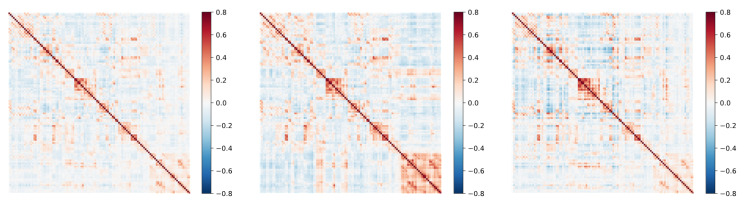
States in HC.

**Figure 9 sensors-25-00156-f009:**
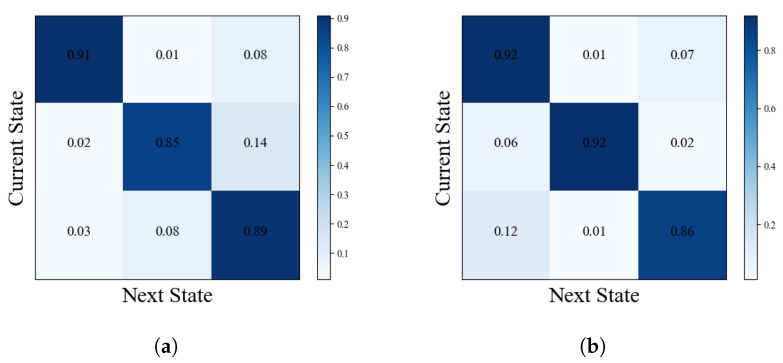
Heat map of state transfer matrix for all subjects. (**a**) ASD. (**b**) HC.

**Figure 10 sensors-25-00156-f010:**
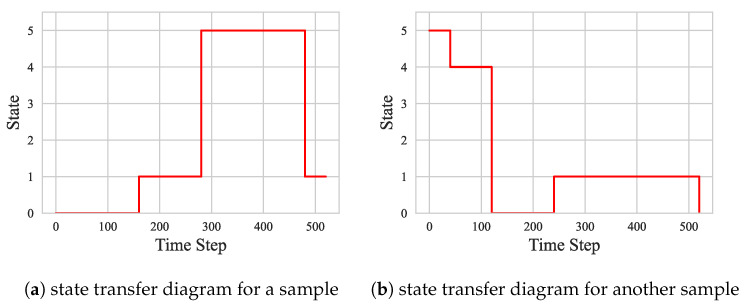
ASD subject state transition diagram.

**Figure 11 sensors-25-00156-f011:**
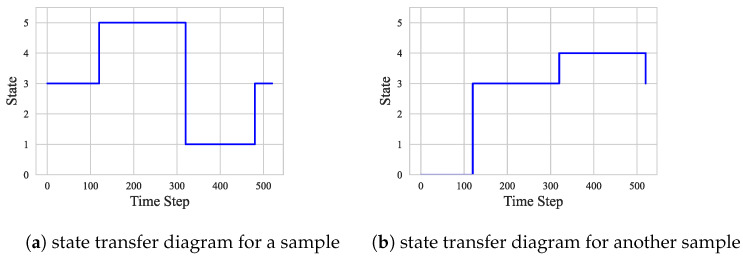
HC subject state transition diagram.

**Figure 12 sensors-25-00156-f012:**
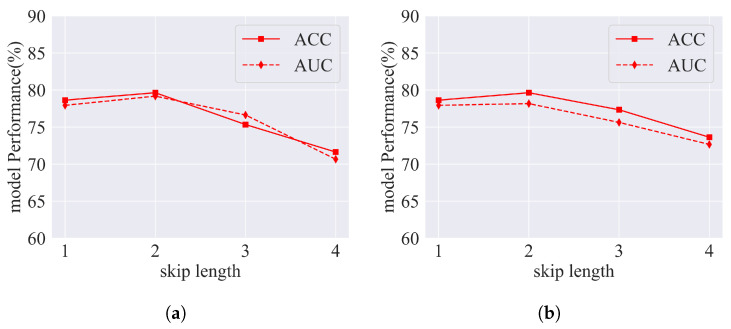
Performance of the model for different *p*. (**a**) ABIDE I. (**b**) ABIDE II.

**Figure 13 sensors-25-00156-f013:**
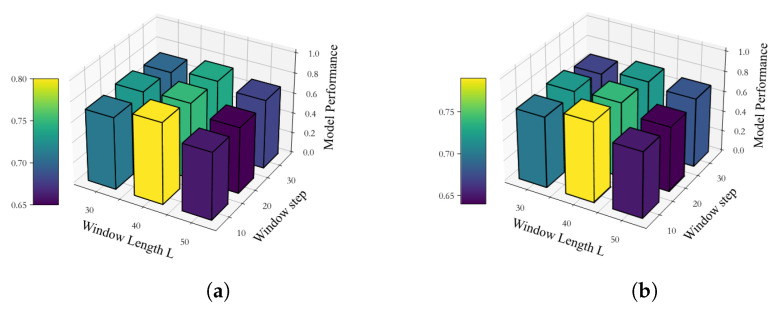
Performance for different window sizes and window step. (**a**) ABIDE I. (**b**) ABIDE II.

**Table 1 sensors-25-00156-t001:** Experimental hardware specifications.

Category	Specifications
System	Ubuntu 20.04.6 LTS (Linux 5.4.0-144-generic)
CPU	16 vCPU AMD EPYC 9654 96-Core Processor
Memory	60 GB
GPU	NVIDIA GeForce RTX 4090 (24 GB)

The CPU is manufactured by the Advanced Micro Devices (AMD), Santa Clara, CA, USA. The GPU is manufactured by the NVIDIA Corporation, Santa Clara, CA, USA.

**Table 2 sensors-25-00156-t002:** Experimental parameter settings.

Parameter	Value
Epoch	60
Batch size	16
Window size	40
Window step size	10
Skip length p	2
Learning rate	0.001
Weight decay	0.05
Scheduler step size	20
Scheduler shrinking rate	0.4
Optimizer	Adam

**Table 3 sensors-25-00156-t003:** Results of different methods on ABIDE I dataset.

Methods	FC Calculation	FC Type	ACC	SEN
Jung 2019 [16]	Pearson correlation coefficient	SFC	0.763	0.792
Liu 2020 [3]	Pearson correlation coefficient	SFC	0.768	0.725
Zheng 2019 [17]	Clustering coefficient	SFC	0.7863	0.8
Zhao 2020 [2]	Central moment value	DFC	0.81	0.82
Kang 2023 [18]	Pearson correlation coefficient	DFC	0.811	NA
Yang 2023 [19]	Pearson correlation coefficient	SFC	0.7874	0.7429
Wang 2024 [20]	Generate interaction of FC	SFC	0.8066	NA
GNN-LSTM	Ledoit–Wolf covariance	DFC	0.804	0.824

**Table 4 sensors-25-00156-t004:** Results of different methods on ABIDE II dataset.

Methods	FC Calculation	FC Type	ACC	SEN
Deng 2023 [9]	Pearson correlation coefficient	DFC	0.7026	NA
Ji 2024 [21]	Pearson correlation coefficient	DFC	0.7181	0.7392
Liu 2023 [22]	Pearson correlation coefficient	DFC	0.72	NA
Hu 2023 [6]	Pearson correlation coefficient	DFC	0.7262	NA
Yang 2023 [19]	Pearson correlation coefficient	DFC	0.8036	0.7624
GNN-LSTM	Ledoit–Wolf covariance	DFC	0.7963	0.7848

**Table 5 sensors-25-00156-t005:** Performance of different methods on ABIDE I.

Methods	ACC	SEN	PRE	F1_Score
GCN	0.6957	0.6522	0.7143	0.6818
LSTM	0.674	0.4783	0.7857	0.5946
GNN-LSTM	0.7391	0.5789	0.7333	0.647
GNN-LSTM (skip)	0.761	0.6667	0.7778	0.7179
GNN-LSTM (pool)	0.7826	0.7647	0.6842	0.7222
GNN-LSTM (skip, pool)	0.8044	0.6111	0.8462	0.7097

skip: denotes the inclusion of jump connections in the model. pool: denotes the use of DG-Pool in the model.

**Table 6 sensors-25-00156-t006:** Performance of different methods on ABIDE II.

Methods	ACC	SEN	PRE	F1_Score
GCN	0.6774	0.6468	0.7026	0.6735
LSTM	0.674	0.6808	0.7056	0.693
GNN-LSTM	0.7278	0.6957	0.7333	0.714
GNN-LSTM (skip)	0.7541	0.6911	0.7458	0.7174
GNN-LSTM (pool)	0.7855	0.7589	0.6998	0.7282
GNN-LSTM (skip, pool)	0.7963	0.7848	0.7762	0.7805

## Data Availability

In this research, a public dataset was used, which can be found at: https://fcon_1000.projects.nitrc.org/indi/abide/, accessed on 24 December 2024.

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
