# Peer review of "Diagnosis of Autism Spectrum Disorder (ASD) by Dynamic Functional Connectivity Using GNN-LSTM"

_sensors, 2024, doi:10.3390/s25010156_

Round 1
Reviewer 1 Report
Comments and Suggestions for Authors
This paper, entitled *"Diagnosis of Autism Spectrum Disorder (ASD) by Dynamic Functional Connectivity Using GNN-LSTM," explores a method for ASD diagnosis by leveraging a Graph Neural Network-Long Short-Term Memory (GNN-LSTM) model. These are my observations:
Line 7-10: How does the proposed GNN-LSTM approach justify its added complexity over simpler models that have previously achieved high ASD classification accuracy? Could a simpler approach achieve similar results without the complex architecture?
Line 38-44: Does the reliance on dynamic functional connectivity (DFC) truly offer a significant advantage over static functional connectivity (SFC) in ASD detection, or is it potentially adding unnecessary complexity without clear clinical benefits?
Line 123-126: Could the jump connection mechanism introduced in the GNN-LSTM model create potential overfitting risks by enabling the model to "memorize" rather than generalize patterns within the DFC data?
Line 286-288: The exclusion of samples that differ in length for consistent data comparison raises concerns about data representativeness. Is there a risk that this standardization excludes important ASD samples, potentially impacting the model's generalizability?
Line 300-306: Given the use of early stopping to avoid overfitting, could this also prematurely stop training, potentially limiting the model's learning and optimization in cases where additional training might improve generalization?
Line 319-326: The high sensitivity and precision focus on ASD detection but lack in-depth discussion on false positive rates. Could this lack of attention to specificity lead to inflated perceptions of the model's accuracy, with implications for clinical applicability?
Line 445-454: The authors suggest future integration of genetic and environmental data. Given that ASD diagnostic challenges persist largely due to biological complexity, is this a realistic goal for improving ASD classification or does it merely shift the focus without addressing the fundamental limitations of fMRI-based approaches?
Add a detailed discussion on the clinical advantages of using DFC over SFC, especially with regard to interpretability and real-world applicability in ASD diagnosis.
Discuss the feasibility of integrating multi-modal data (e.g., genetic, physiological) with the GNN-LSTM model in terms of data availability, processing challenges, and realistic expectations for impact on ASD diagnosis.
Address the potential limitations and bias introduced by excluding data of variable lengths, explaining how it could impact generalizability or representativeness for real-world applications.
Reviewer 2 Report
Comments and Suggestions for Authors
In the abstract you write:
" The model achieves remarkable results on the ABIDE dataset with accuracies of 80.4% and 79.63%, respectively,”
Specify what the two reported accuracies are, so it is not clear, do you mean performance on “the ABIDE I and ABIDE II datasets”?
If possible, upload the code to github, it is useful to the community.
Explain the datasets better; it is not clear how they were constructed, how many types of instrumentation were used? how many clinical centers? and so on; it is important to understand whether the results are realistic compared to the real world, e.g., whether training is done on different clinics from those used in the test set, and so on
The main concern is that you must motivate your proposed architecture against other architectures, based on graph neural net and lstm combination, proposed in the literature, e.g., compare your architecture with:
https://www.sciencedirect.com/science/article/pii/S0141029624002955
https://link.springer.com/chapter/10.1007/978-3-031-09342-5_11
“the dataset was randomly partitioned into a training set and a test set based on the standard 8:2 ratio.”
For each method you compare your method with, you have to specify the test protocol, I have tried to check other papers and they use different protocols. You cannot compare your method with other approaches if they use a different testing protocol than your, if you use a 5-fold cross validation you cannot compare your results with those obtained using a 10-fold cross validation. I suggest to use a test protocol used in the literature and compare only with papers that use exactly the same protocol.
Round 2
Reviewer 1 Report
Comments and Suggestions for Authors
The observations were corrected. Thank you.
Reviewer 2 Report
Comments and Suggestions for Authors
revision well done